# Sex differences in the immediate extinction deficit and renewal of extinguished fear in rats

**Annalise N. Binette**[☉], **Michael S. Totty**[iD][☉], **Stephen Maren**[iD]*

Department of Psychological and Brain Sciences and Institute for Neuroscience, Texas A&M University, College Station, Texas, United States of America

☉ These authors contributed equally to this work.
* maren@tamu.edu

**Data Availability Statement:** All relevant data are within the paper and its Supporting Information files.

## Abstract

Extinction learning is central to exposure-based behavioral therapies for reducing fear and anxiety in humans. However, patients with fear and anxiety disorders are often resistant to extinction. Moreover, trauma and stress-related disorders are highly prone to relapse and are twice as likely to occur in females compared to males, suggesting that females may be more susceptible to extinction deficits and fear relapse phenomena. In this report, we tested this hypothesis by examining sex differences in a stress-induced extinction learning impairment, the immediate extinction deficit (IED), and renewal, a common form of fear relapse. In contrast to our hypothesis, there were no sex differences in the magnitude of the immediate extinction deficit in two different rat strains (Long-Evans and Wistar). However, we did observe a sex difference in the renewal of fear when the extinguished conditioned stimulus was presented outside the extinction context. Male Wistar rats exhibited significantly greater renewal than female rats, a sex difference that has previously been reported after appetitive extinction. Collectively, these data reveal that stress-induced extinction impairments are similar in male and female rats, though the context-dependence of extinction is more pronounced in males.

## Introduction

Clinical disorders caused by trauma exposure (e.g., post-traumatic stress disorder, PTSD) afflict millions of men and women in the United States every year [1]. Importantly, women are twice as likely as men to develop PTSD, suggesting that biological sex may influence the neural and behavioral processes underlying the pathophysiology of stress- and trauma-related disorders [2]. Consistent with this, a substantial body of work has revealed sex differences in learning and memory processes that may contribute to the development and maintenance of PTSD. For example, there are robust sex differences in Pavlovian fear conditioning in rats [3–6]. In this form of learning, a neutral conditioned stimulus (CS), such as an acoustic tone, is arranged to precede and predict a noxious unconditioned stimulus (US), such as an electric footshock. After as little as a single conditioning trial, presentation of the CS (or placement in the

**Funding:** The work was supported by grants from the National Institutes of Health (R01MH065961 and R01MH117852) to SM.

**Competing interests:** The authors have declared that no competing interests exist.

conditioning context) elicits anticipatory, conditioned fear responses (CRs), including freezing behavior [7–10]. Several studies have revealed that male rats exhibit greater acquisition of conditioned freezing to the environmental stimuli associated with shock (i.e., the conditioning context) relative to females [5,11–16], whereas freezing to discrete auditory CSs is similar in males and females [5,16,17]. Interestingly, there are also sex differences in hippocampal and amygdala synaptic plasticity mechanisms thought to underlie these forms of learning [5,18]. These results suggest that sex differences in the neural and behavioral mechanisms of aversive learning and memory may contribute to the pathophysiology of PTSD.

Another form of learning implicated in the maintenance of PTSD is extinction. Extinction is a form of learning in which repeated non-reinforced presentations of a conditioned stimulus (CS) reduce the magnitude reduction of the CRs acquired during Pavlovian conditioning [19,20]. Importantly, extinction is central to cognitive-behavioral therapies for PTSD, including prolonged exposure therapy [21]. In prolonged exposure therapy, trauma-related stimuli are repeatedly presented within a safe setting until those stimuli no longer elicit fear. Although extinction-based therapy can be highly effective at reducing pathological fear, the durability of extinction memory can be compromised by a variety of factors [22,23]. For instance, suppression of a CR after extinction is typically limited to the setting or context in which extinction learning occurred, resulting in renewal of the CR outside of the extinction context [24]. Furthermore, stress (acute or chronic) can considerably reduce extinction learning and long-term retention, thereby fear relapse [25]. For example, the "immediate extinction deficit" (IED) occurs when extinction training is conducted shortly after fear conditioning (a stressor), rendering impairments in long-term extinction memory and relapse of conditioned fear [26,27]. Interestingly, immediate extinction occurs against the background of high levels of generalized contextual fear, which has been proposed to play an important role in the IED [27] (though see [28]).

Although extinction learning plays a central role in behavioral therapies for trauma- and stressor-related disorders, the interaction of sex differences in contextual fear conditioning, on the one hand, with extinction learning and memory, on the other, is unclear. In addition to sex differences in contextual fear conditioning, there is considerable evidence that stress-related neural circuitry in females are more sensitive to those in males [29–32]. These sex differences might make females more susceptible to stress-induced extinction learning impairments, such as the IED. Moreover, sex differences in contextual conditioning and stress responsivity might influence the relapse of extinguished fear, including renewal. Therefore, in this study, we characterized the IED in male and female rats using two common laboratory rat strains (Long Evans and Wistar) and standard Pavlovian fear conditioning and extinction procedures. Surprisingly, we found that there was no sex difference in the IED in either rat strain; both male and female rats exhibited poor extinction retention relative to animals undergoing delayed extinction and exhibited similar levels of conditioned freezing compared with non-extinguished controls. In contrast, male Wistar rats showed greater renewal of fear to an extinguished CS in a novel context compared to females. Sex differences in renewal were not due to greater contextual fear in male rats. These results reveal that male and female rats are similarly susceptible to stress-induced extinction impairments, but that males may be more susceptible to fear renewal.

## Materials and methods

### Ethics statement

This study was carried out in strict accordance with the recommendations of the National Institutes of Health and Texas A&M University. These experiments were approved by the

Texas A&M University Animal Care and Use Committee (Animal Use Protocol Number: 2020–0305).

## Subjects

A total of 64 adult female and male Long-Evans and Wistar rats were used in this study. Long-Evans Blue Spruce rats (male, n = 16; female, n = 16) were obtained from Envigo (Indianapolis, IN) and weighed 200–224 g upon arrival. Wistar rats (male, n = 16; female, n = 16) were bred in-house; they were 11–16 weeks of age upon behavioral testing. The Wistar rats used in this study were derived from *Crh*-Cre rats [33] obtained from the laboratory of Robert Messing at the University of Texas at Austin. These rats were bred in our laboratory with commercially supplied wild-type Wistar rats (Envigo, Indianapolis, IN). The Wistar rats were genotyped and randomly assigned to groups (extinction type) within each litter; both Cre$^+$ and Cre$^-$ rats were included in the experiments and their behavior was identical. All rats were individually housed in a temperature- and humidity-controlled vivarium, with a 14:10 hour light/dark cycle and ad libitum access to food and water. Behavioral testing was conducted during the light phase. Rats were handled 1 minute per day for 5 days prior to testing to acclimate them to the experimenter.

## Procedure

All behavioral procedures took place in a standard rodent conditioning chamber with two aluminum walls, two Plexiglas walls and a Plexiglas ceiling (Med Associates, St. Albans, VT). The chamber was outfitted with a speaker mounted to the upper corner of one wall for delivery of auditory stimuli. The grid floor was composed of stainless-steel rods for delivery of scrambled footshock. Load-cell force transducers located underneath each chamber measured displacement of the chamber in response to motor activity; these voltages (+/-10 V) were acquired at 5 Hz and transformed to absolute values (scale of 0–100). A freezing bout was defined as five consecutive values below 10 (freezing threshold, corresponding to one second of freezing).

To assess the IED, we conducted separate experiments in Long-Evans and Wistar rats. In the first experiment, male and female Long-Evans rats underwent either immediate or delayed extinction after auditory fear conditioning. Extinction was followed by a retention test conducted 24 hours later. All the behavioral procedures were conducted in the same context (Context A). Animals were transported from the vivarium to the laboratory in black plastic boxes. A metal pan beneath the grid floor of each conditioning chamber was cleaned with a 1% ammonium solution. The room housing the conditioning chambers was illuminated with red light and fans affixed to each chamber were turned on. Doors of the outer sound attenuating cabinets were closed. Conditioning consisted of a 3-min baseline period followed by 5 CS-US pairings. The CS was a 10-s white noise (80 dB) paired with a 2-s, 1-mA footshock unconditioned stimulus (US). The intertrial interval (ITI) was 70 seconds. After conditioning, animals were placed in the transport boxes and returned to the vivarium unless undergoing immediate extinction, in which case they remained in the transport boxes until the immediate extinction session began. Either 15-min or 24-hr later, the animals were returned to the conditioning chambers for immediate or delayed extinction, respectively. Extinction consisted of a 3-min baseline period followed by 45 CS-alone presentations (40-s ITI). All animals were returned to the chamber again 48-hr after conditioning for a retrieval test consisting of a second extinction session.

In the second experiment, we examined the IED in Wistar rats. Because the IED has not previously been studied in Wistar rats, we used a no-extinction control similar to our previously published work in Long-Evans rats [26]. The conditioning procedures were identical to

those used for the Long-Evans rats, except for the use of a stronger footshock US (2 mA). After conditioning, animals received either an immediate or delayed extinction procedure or were simply placed in the conditioning chambers (no-extinction control); extinction retrieval testing was conducted 48 hours after. Conditioning and extinction procedures were conducted in Context A. After the retrieval test, all animals underwent a second extinction session (re-extinction) then were randomly assigned to a retrieval test (5 CS-only presentations) in either Context A (SAME, the extinction context) or Context B (DIFF, a novel context). Animals tested in Context B were transported from the vivarium to the laboratory in white transport boxes. For Context B, the conditioning chambers had a metal pan beneath the grid floor that was cleaned with a 3% acetic acid solution, and the room was illuminated with standard white light. Fans affixed to the sound attenuating cabinets were turned off and doors to the cabinets were left open.

## Statistics

Data were analyzed with StatView software (SAS Institute). One female Long-Evans rat failed to acquire fear during conditioning due to technical difficulties and was thus excluded from statistical analyses. Results are displayed as mean ± standard error the mean (SEM). Analysis of variance (ANOVA) was used to assess percentage of time freezing with repeated measures of trial ($\alpha$ = 0.05).

## Results

### Both male and female rats display the immediate extinction deficit

We first sought to determine if both male and female rats are similarly susceptible to the IED, a stress-induced extinction impairment (Fig 1A). To this end, Long-Evans rats underwent standard auditory fear conditioning which consisted of 5 CS-footshock (US) pairings. All rats acquired fear to the CS [main effect of Trials: $F_{5, 135} = 99.8$, $p < .0001$] and there were no differences between males and females [no main effect of Sex: $F_{1, 27} = 0.60$, $p = 0.45$; Trials x Sex interaction: $F_{5, 135} = 0.46$, $p = .81$] (Fig 1B; Conditioning). Rats next underwent fear extinction either 15 minutes (Immediate) or 24 hours (Delayed) later in the same context, which consisted of 45 CS-alone trials. We found that rats extinguished immediately following fear conditioning showed impaired within-session extinction compared to animals extinguished 24 hours later [main effect of Ext Type: $F_{1, 27} = 6.21$, $p = 0.02$; Block x Ext Type interaction: $F_{9, 243} = 2.87$, $p = 0.003$] (Fig 1B; Extinction). Forty-eight hours after conditioning, all animals were brought back to the extinction context and tested for extinction memory with another 45 CS-alone trials. During retrieval, both groups froze at similar levels to the first five trials, however, following this the Delayed animals reduced their freezing, demonstrating good extinction memory, while high freezing in the Immediate group persisted [Block x Ext Type interaction: $F_{8, 216} = 2.90$, $p = .004$] (Fig 1B; Retrieval). Hence, the IED was manifest as greater re-extinction (savings) in the animals that underwent delayed extinction compared to those undergoing immediate extinction as we have previously reported [34]. Importantly, although females showed a slightly faster reduction in freezing compared to males [$F_{8, 216} = 1.72$, $p = 0.09$], the immediate extinction procedure produced a similar deficit in extinction retention in both male and female rats [Block x Ext Type x Sex interaction: $F_{8, 216} = 0.274$, $p = 0.97$]. These data suggest that male and female Long-Evans rats are similarly susceptible to the immediate extinction deficit.

There are some inconsistencies in the reported findings of sex differences in learning and memory paradigms and it has been suggested that some of these findings may be strain specific [35]. We thus sought to replicate the above findings using a different strain of rats, while

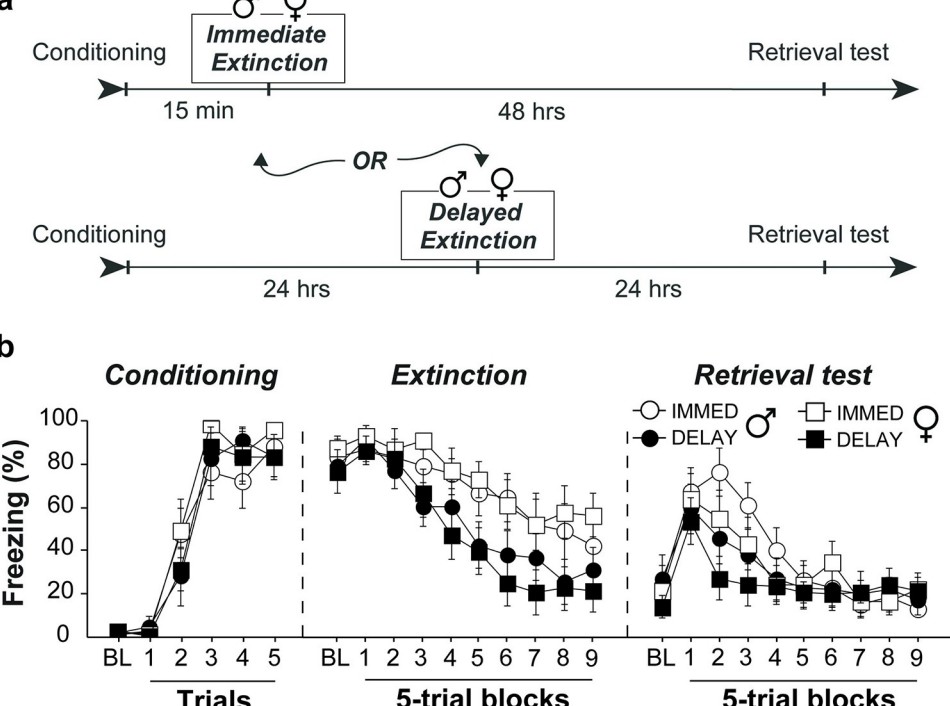

**Fig 1. Male (♂) and female (♀) Long-Evans rats exhibit the IED. (A)** Behavioral timeline for the different experimental groups. **(B)** Freezing data show that Long-Evans rats acquired equivalent levels of conditional fear to the auditory CS (Conditioning). Both Immediate and Delayed groups showed a marked reduction in fear to the CS throughout fear extinction, with Delayed animals showing lower levels of fear at the end of extinction training (Extinction). Although all groups of animals showed equivalent levels of fear early in the retrieval test (Block 1), immediately extinguished animals demonstrated a reduced rate of re-extinction compared to Delayed animals, indicative of impaired extinction memory (Retrieval). This IED was similar between male and females, though, females showed faster rate of re-extinction compared to males in both groups. All error bars represent ± SEM.

adding control groups that do not undergo fear extinction (Fig 2A). Equal numbers of Cre$^+$ and Cre$^-$ male and female Wistar rats (see the *Materials and Methods* section for more detail) were used for the following experiments. Throughout all statistical comparisons, we observed no significant effects of genotype and thus chose to collapse data across genotype to improve statistical power. All rats were first conditioned as previously described and all rats developed conditional responding to the CS [main effect of Trial: $F_{5, 140}$ = 50.24, $p < 0.0001$] (Fig 2B). Female rats showed slightly higher freezing compared to males [main effect of Sex: $F_{1, 28}$ = 8.65, $p < 0.0065$] and, although Immediate and Delayed groups displayed slightly different learning curves [Trial x Ext Type interaction: $F_{5, 140}$ = 5.01, $p = 0.0003$], they displayed equivalent total amounts of freezing [no main effect of Ext Type: $F_{1, 28}$ = 0.02, $p = 0.90$] (Fig 2B). Following conditioning, rats either underwent extinction 15 minutes (Immediate-Ext) or 24 hours (Delayed-Ext) after conditioning, or they were merely re-exposed to the context 15 minutes (Immediate-NoExt) or 24 later hours (Delayed-NoExt) (Fig 2A). Although female rats showed more freezing during conditioning, we did not observe any sex differences during extinction [no main effect of Sex: $F_{1, 24}$ = 0.03, $p = 0.86$; Block x Sex interaction: $F_{9, 216}$ = 0.61, $p = 0.79$; Ext Type x Sex interaction: $F_{1, 24}$ = 0.61, $p = 0.44$; or Block x Ext Type x Sex interaction: $F_{9, 216}$ = 0.76, $p = 0.66$].

It is of note that Wistar rats displayed remarkably low baseline freezing during both the immediate and delayed extinction procedures (< 20%; Fig 2C) compared to Long-Evans rats

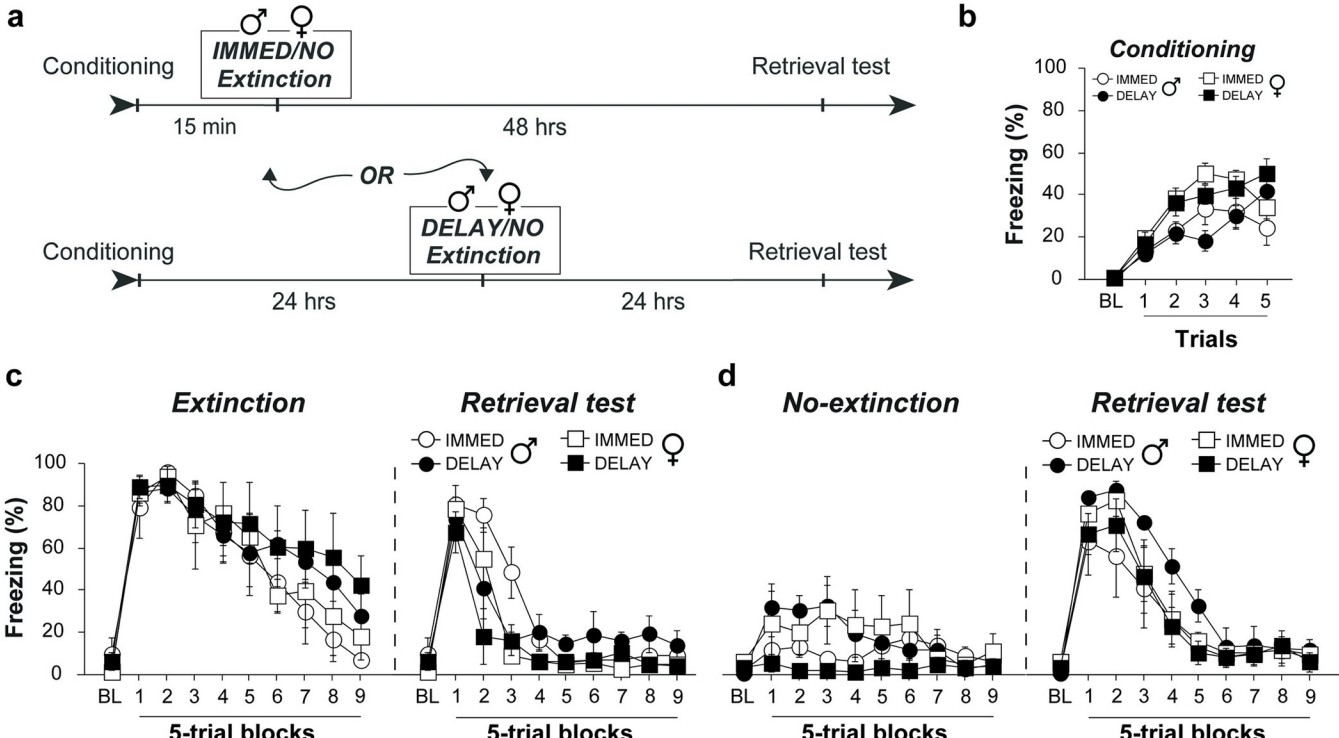

**Fig 2. Male (♂) and female (♀) Wistar rats exhibit an IED. (A)** Behavioral timeline for the different experimental groups. **(B)** All groups acquired similar levels of conditioned freezing prior to extinction. **(C)** Wistar rats that underwent extinction training showed a reduction in fear across extinction trials with immediately extinguished rats reaching lower levels of freezing than delayed animals. Despite this, immediate extinction animals still showed impaired extinction during retrieval testing as indicated by reduced rate of re-extinction. This effect was similar in both males and females. **(D)** Comparatively, No-Extinction animals exhibited high levels of conditioned freezing during retrieval testing that was similar among all groups. All error bars represent ± SEM.

in the previous experiment (~80%; Fig 1B). This appears to be a strain difference as similarly low levels of contextual fear have previously been reported in both Wistar [36] and Sprague-Dawley rats [37]. Nonetheless, Wistar rats exhibited high levels of CS-evoked freezing at the outset of extinction training compared to control animals that were merely exposed to the context [main effect of Ext: $F_{1, 24} = 68.38$, $p < 0.0001$; Block x Ext interaction: $F_{9, 216} = 17.21$, $p < 0.0001$] (Fig 2C and 2D). We additionally observed higher levels of contextual fear in male rats that were exposed to the conditioning context 24 hours after conditioning [Block x Sex x Ext-type interaction: $F_{9, 108} = 2.562$, $p = 0.0104$; Fischer's PLSD: $p = .036$].

Like the previous experiment, all groups showed equally high freezing during the first 5 trials of retrieval testing; however, delayed extinction animals exhibited a clear decrease in freezing in the second block of 5 trials whereas high freezing persisted in all other groups [Block x Ext x Ext Type interaction: $F_{8,192} = 3.50$, $p = 0.0009$] (Fig 2C and 2D). This effectively demonstrates that undergoing extinction training immediately following conditioning impairs extinction retention to that of animals that never underwent extinction at all. Importantly, we once again show no sex differences in the IED when comparing across extinction groups [no Block x Ext x Ext Type x Sex interaction: $F_{8,192} = 1.88$, $p = 0.065$] or in any other comparison during retrieval testing (all $p$-values > .09). Collectively, we show that male and female rats, across strains, are equally susceptible to stress-induced extinction impairments.

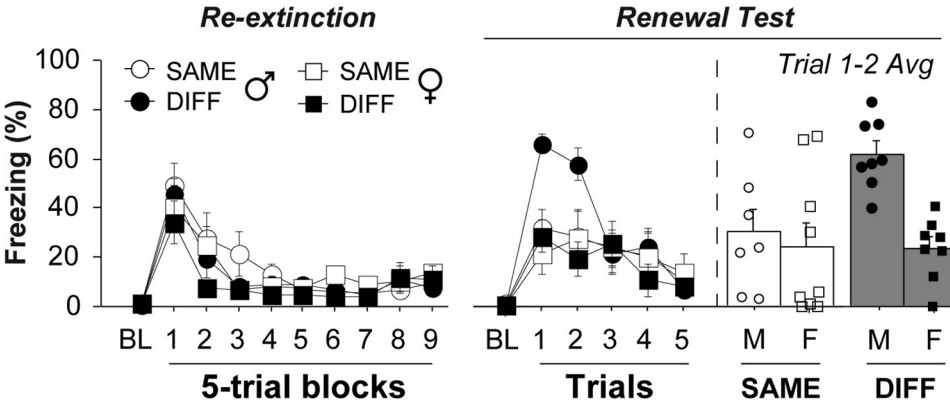

**Fig 3. Male (♂) but not female (♀) Wistar rats exhibit renewal of extinguished fear.** Freezing data showing that male and female Wistar rats displayed similar levels of fear during a second extinction session after the prior extinction retrieval test (left). However, only male rats displayed fear renewal the following day when presented the extinguished CS in a novel context (DIFF, right) relative to animals tested in the extinction context (SAME). This was particularly evident during the first two trials of testing. All error bars represent ± SEM.

## Male, but not female, rats exhibit renewal of extinguished fear

Similar to stress-induced impairments in extinction learning, contextual processing is thought to be central to trauma-related disorders such as PTSD [38,39]. Previous work has demonstrated sex differences in the renewal of extinguished CRs after appetitive conditioning [40], however, it is currently unknown if this is true for aversively conditioned CSs. To test this, Wistar rats from the previous experiment underwent an additional extinction session to completely extinguish any remaining fear (Fig 3). The animals were then reassigned to new groups for renewal testing and were counterbalanced for previous group assignments. After the second extinction session, male and female rats were either placed back into the extinction context (A; retrieval context) or they were placed into a novel context (B; renewal context) where they were presented 5 CS-alone trials. During the second extinction session all rats showed moderate levels of fear during the first five-trial block but quickly reduced freezing to baseline levels [main effect of trials: $F_{9,252} = 34.09$, $p < 0.0001$], and there were no sex or group differences during extinction [Trials x Sex interaction: $F_{9,252} = 1.579$, $p = 0.12$; Trials x Context interaction: $F_{9,252} = 1.169$, $p = 0.32$; Trials x Sex x Context interaction: $F_{9,252} = 0.654$, $p = 0.75$] (Fig 3; Extinction II). During testing, both male and female rats placed back into the extinction context showed very low levels of freezing. However, males, but not females, placed into the novel context showed a renewal of conditioned freezing [Trials x Sex x Context interaction: $F_{5,140} = 2.60$, $p = 0.028$], particularly during the first two trials (Fig 3; Renewal Test). Importantly, these results cannot be explained by prior extinction group assignments as an analysis of the renewal data including this variable did not reveal significant Sex x Ext (all $p > 0.4$) or Sex by Ext-type interactions (all $p > 0.3$). These data demonstrate that male rats are more susceptible to the renewal of extinguished fear.

## Discussion

In this study we investigated sex differences in failures of extinction learning and retrieval. We found that male and female rats are equally susceptible to the IED—a stress-induced extinction impairment—in two different strains of animals. Interestingly, we also show that females, but not males, exhibit impairments in the renewal of extinguished fear, suggestive of a sex difference in contextual processing. This finding is in line with previous work showing sex

differences in the renewal of appetitive conditioning [40]. Our work extends previous literature and suggests that differences in contextual processing may be a critical factor accounting for the disparity in susceptibility for stress and trauma-related disorders across sexes.

Although both male and female rats show comparable deficits in immediate extinction, both male and female rats showed a more rapid (but non-significant) reduction in conditioned freezing during retrieval testing, regardless of the extinction procedure (immediate or delayed). It is unlikely that this rate difference is a consequence of sex differences in conditioning insofar as Long-Evans rats showed no sex differences in conditioning, whereas Wistar females showed greater levels of freezing during conditioning. A potential explanation for sex differences in the (re)extinction of fear is estrous cycle phase. For example, extinction is impaired in females that undergo extinction training during metestrus but enhanced in females trained in estrus or proestrus [6,41]. Ovarian steroids underlie the more rapid reduction in freezing during extinction retrieval in females [42,43], which was observed under both low- and high-stress conditions. This suggests that stress does not block the adaptive effects of high estrogen and progesterone on retrieval facilitation. From another perspective, however, changes in hormonal state across extinction training and retrieval testing in females might be expected to impair extinction retrieval and facilitate renewal because extinction retrieval is highly context-dependent [24]. Yet the opposite was true: female rats showed superior extinction retrieval and did not exhibit renewal (at least in Wistar rats). It would be important in future work to examine the role that estrous cycle and gonadal steroids may play in these processes.

Previous work has found that female rats display lower levels of contextual fear after auditory or contextual fear conditioning [5,11–16]. In the present study, we did not observe sex differences in contextual freezing during the baseline periods of extinction sessions, which typically serves as a measure of contextual fear. However, in the immediate extinction procedure this baseline period is confounded by non-associative sensitization that summates with associative fear to the context [44]. In addition, contextual fear during the baseline period of the extinction retrieval test is confounded by context extinction (or lack thereof) experienced during the extinction training session. In Experiment 2, however, rats in the Delay-No-Extinction group were placed into the conditioning context (24 h after conditioning) for a ~35 min test with no CSs. In this test of contextual freezing, which was not confounded by sensitization, male Wistar rats exhibited reliably more freezing than female rats.

Given that the IED is a stress-induced deficit driven by the locus coeruleus (LC) [27,45,46], a sexually-dimorphic structure [29], it is perhaps surprising that we did not observe sex differences in the IED. Recent work has revealed that the IED is driven by LC-derived norepinephrine (LC-NE) which excites neurons in the basolateral nucleus of the amygdala (BLA) [45,47,48] and, in turn, impairs the infralimbic (IL) division of the medial prefrontal cortex [49–51]. The IL is critical to the formation of long-term extinction memories [52–54]. Although LC-NE release within the prefrontal cortex enhances arousal and is necessary for successful extinction learning [55], high levels of LC-NE may act to impair prefrontal function [45,56]. Substantial work has demonstrated that LC neurons in females compared to males are genetically distinct [57], have greater dendritic morphology [58], and are 10-30x more sensitive to activation by corticotropin-releasing factor (CRF), irrespective of cycling hormones [59]. Moreover, the central nucleus of the amygdala (CeA) is a large source of CRF input that is directly excitatory to LC-NE neurons [60–64], and CeA CRF+ cells have recently been shown to be necessary and sufficient to drive the IED [65]. All of this would suggest that females should be highly susceptible to stress-induced extinction impairments. Nonetheless, it has also been found that prior stress sensitizes LC neurons in male rats [66,67], but not females, which abolishes sex differences in LC sensitivity [59]. Thus, we suggest that high levels

of physiological stress (such as that from fear conditioning) may result in a ceiling effect such that male and female rats are equally susceptible to stress-induced impairments in extinction learning. Given this, it would be worthwhile to know if female rats are susceptible to the IED under conditions that are typically not stressful enough to cause extinction impairments in male rats, such as weaker shocks or fewer conditioning trials [45,68]. Although acute stress from fear conditioning in rodents does not appear to result in obvious sex differences in the IED procedure, it is still possible that the wide-range of sex differences in the LC-NE system plays a central role in the development of chronic stress and/or psychiatric disorder, such as PTSD, in humans.

Our finding that male, but not female, rats exhibit renewal of extinguished fear mirrors work in both appetitive and aversive conditioning [40,69]. Anderson and Petrovich (2015) investigated the renewal of food seeking using an ABA design in Long-Evans rats and found that males, but not females, renewed previously extinguished conditional responding to the food cup when tested outside of the extinction context [40]. They additionally found that renewal returned in ovariectomized female rats that received estradiol replacement throughout behavioral training and testing, but not in ovariectomized females without estradiol replacement. This suggests that levels of estradiol may play a critical role in renewal in female rats [40]. Contrary to this, Hilz and colleagues have shown that renewal in females only occurs when they are extinguished in the proestrous phase (high levels of estrogen) and tested in metestrous/diestrous phase (low levels of estrogen), suggesting that renewal in intact females relies on a shift in interoceptive hormonal state [70,71]. Further adding to this theory, Park and colleagues show that female rats display renewal at a juvenile age (P18) before hormone cycling begins [72,73]. In aversive conditioning, estrous cycle phase has been shown to modulate fear renewal in female rats. Female rats that undergo auditory fear extinction during metestrus and diestrus, but not in proestrus and estrus, show fear renewal to the CS in a novel context [69]. Taken together, we suggest that renewal in female rats depends heavily on both interoceptive and exteroceptive context.

Although we failed to observe renewal in female rats, previous reports have observed robust renewal in females [24,74–76]. Indeed, Bouton and Bolles' seminal findings found renewal in female Wistar rats [24]. Notably, however, they measured fear using a conditioned suppression procedure in which rats show a reduction in lever pressing for food during an aversive CS. Response competition in this test situation may influence sex difference in renewal. Nonetheless, others have shown that female rats exhibit renewal using freezing as the dependent variable [75,76], suggesting that this discrepancy is unlikely to be due to procedural differences. The higher levels of contextual freezing observed in the present experiment in male rats suggests that exteroceptive stimuli may have stronger control of their behavior, including the context-dependence of extinction. Collectively, there is still much work to be done to determine the factors that regulate sex differences in fear relapse after extinction.

Indeed, sex differences in brain areas critical for the context-dependence of extinction may underlie differences in renewal. The hippocampal formation encodes and transmits spatial information to various limbic regions, including the mPFC and amygdala, primarily through its ventral subregion (vHPC) that is critical for guiding contextually appropriate behaviors. The vHPC sends monosynaptic projections to the lateral and central nuclei of the amygdala [77,78], and a dense feedforward inhibitory circuit to the mPFC [79,80], each of which have been implicated in context-dependent fear renewal. Appetitive renewal similarly involves the activation of mPFC-projecting vHPC neurons [81], modulation of the mPFC [82,83], and amygdala activation [70,71,83]. Interestingly, sex differences in the recruitment of these neural circuits mirrored the behavioral sex differences seen in appetitive renewal [70,71,81–83]. It

seems likely that the failure to recruit these circuits when confronted with an extinguished CS outside of the extinction context underlies impaired renewal often observed in female rats.

In summary, we first show that both male and female rats are equally susceptible to the IED despite well-known sex differences in the neural circuits underlying the IED. We speculate that enhanced basal excitability of the LC-NE system in females may result in increased susceptibility to mild stressors compared to male rats, whereas strong acute stressors result in the sensitization of the male LC-NE system, abolishing potential sex differences in extinction learning. We next show that female rats do not display context-mediated fear renewal, similar to reports in appetitive literature. We argue that renewal of conditional responding in female rats may depend heavily on interoceptive, in addition to exteroceptive, contexts. Changes in interoceptive context are monitored by the insular cortex [84], and it's possible that insular projections to the amygdala and para-hippocampal regions may gate the renewal of extinguished fear. Alternatively, cycling hormones such as estradiol may modulate the excitability of limbic circuits underlying renewal.

## Supporting information

**S1 Data.**
(XLSX)

## Author Contributions

**Conceptualization:** Annalise N. Binette, Michael S. Totty, Stephen Maren.

**Data curation:** Annalise N. Binette, Michael S. Totty, Stephen Maren.

**Formal analysis:** Annalise N. Binette, Michael S. Totty, Stephen Maren.

**Funding acquisition:** Stephen Maren.

**Investigation:** Annalise N. Binette, Michael S. Totty.

**Methodology:** Annalise N. Binette, Stephen Maren.

**Supervision:** Stephen Maren.

**Visualization:** Stephen Maren.

**Writing – original draft:** Annalise N. Binette, Michael S. Totty.

**Writing – review & editing:** Annalise N. Binette, Michael S. Totty, Stephen Maren.

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
