## [Decision Letter · Decision Letter 0]

8 Mar 2022

PONE-D-22-04660Sex differences in the immediate extinction deficit and renewal of extinguished fear in ratsPLOS ONE

Dear Dr. Maren,

Thank you for submitting your manuscript to PLOS ONE. After careful consideration, we feel that it has merit but does not fully meet PLOS ONE’s publication criteria as it currently stands. Therefore, we invite you to submit a revised version of the manuscript that addresses the points raised during the review process.

We look forward to receiving your revised manuscript.

Kind regards,

Fred J. Helmstetter, PhD

Academic Editor

PLOS ONE

Journal Requirements:

Reviewers' comments:

Reviewer's Responses to Questions

**Comments to the Author**

1. Is the manuscript technically sound, and do the data support the conclusions?

Reviewer #1: Partly

Reviewer #2: Partly

2. Has the statistical analysis been performed appropriately and rigorously? 

Reviewer #1: I Don't Know

Reviewer #2: Yes

3. Have the authors made all data underlying the findings in their manuscript fully available?

Reviewer #1: Yes

Reviewer #2: Yes

4. Is the manuscript presented in an intelligible fashion and written in standard English?

Reviewer #1: Yes

Reviewer #2: Yes

5. Review Comments to the Author

Reviewer #1: This manuscript examined whether or not biological sex has an effect on the retrieval of extinction memory when rats are exposed to extinction training within minutes of fear conditioning versus extinction training given 24 hours after conditioning. The authors also tested whether behavioral performance differed in males in females given a renewal test after extinction. The manuscript is well written and the experiments are well-designed. I have some comments related to analysis and interpretation that I thing might be useful to address.

1) I wonder if the authors can speak to whether or not the deficit in extinction when it occurs shortly after conditioning might not be an effect on extinction per se, but instead reflects enhanced conditioning that is not evident on the early trials of extinction because performance is at ceiling.

2) Experiments 1 and 2 appear to be analyzed differently with experiment 1 being analyzed with trials, extinction type, and sex as factors, while experiment 2 used blocks instead of trials. I wonder if the authors could explain why the difference.

3) Since there was no context shift, the baseline periods of the extinction training session functions as a test of context fear. The authors should comment on the lack of a difference between males in females (which contradicts some prior work) and how it fits with their explanation of the sex difference in renewal as reflecting a difference in contextual processing. The authors might also comment on why there is such a large discrepancy in experiments 1 and 2 in contextual fear. Is this simply a strain effect?

4) The conclusion that immediate extinction produces less long-term extinction is based on performance during the retrieval test. In both experiments 1 and 2, there is no difference during the first block of 5 trials of the retrieval test, but a difference emerges on the subsequent trials. This is reported as a trial (or block) by extinction type interaction. I wonder why this interaction is taken as evidence of a difference in extinction retention, while the lack of an effect on the first 5 trials is not evidence of a lack of an effect. Especially when considering how common it is in the literature for extinction retention to be assessed by analyzing the initial trials of retrieval, and not the later trials which would be confounded by re-extinction.

Reviewer #2: The present manuscript describes two experiments aiming to examine potential sex differences in the immediate extinction deficit (IED), finding no sex-based differences in two strains of rats. A sub-experiment demonstrates that in Wistar rats, females fail to show AAB renewal following a complete IED experiment and re-extinction. Overall the paper is well-written and the experiments are appropriately conducted and analyzed, but the general conclusion that females fail to show renewal of a fear response is difficult to reconcile with the broader literature. Instead, this effect might be an artifact of the procedure employed here which involved different extinction treatments ahead of the renewal testing.

Renewal of a fear response has been well-documented in female Wistar rats. For example, the Bouton & Bolles (1979) demonstration of fear renewal used exclusively female Wistar rats. One procedural difference between the Bouton & Bolles results and the results reported here is the use of conditioned suppression in the 1979 demonstration and use of freezing here. It should be stated that renewal has also been observed in female Wistar rats when freezing was the dependent variable of interest (e.g., Harris & Westbrook, 1998, Expt. 5; Morris & Bouton, 2007). A failure to obtain fear renewal in the present study was therefore unlikely to be due to a difference in the dependent variable of choice and instead attributable to other factors here.

Most notably, the authors used non-naïve rats who had previously undergone conditioning through an IED paradigm which involved Pavlovian fear conditioning, extinction, and testing. While the authors did report difference between immediate and delayed extinction procedures in later extinction recall, the groups are collapsed for the renewal experiment. I think it would be worth analyzing the data taking into account prior group assignment (immediate vs. delayed) to examine if differences in the formation of the original extinction memory are contributing to the effects observed in later renewal.

An additional caveat to the present results is that rather than employing an ABA renewal design, the authors examined AAB renewal which is a relatively less robust effect. There are a few possible explanations of the results. Either, as the authors state, females are less likely to show AAB renewal than males (which is interesting in its own right as there are several examples of ABA renewal in females) or something about the prior experience with extinction in the IED paradigm renders females less susceptible to AAB renewal.

I hesitate to suggest new experiments because I am acutely aware of the amount of work that goes into running these, but I don’t see how the authors can make concrete claims on the inability for females to demonstrate fear renewal without running additional experiments. I do think that in order to publish this with the current data, a new analysis including groups separated in the renewal test based on prior group assignment as well as discussions highlighting procedural differences here that might contribute to the lack of renewal, especially in the context of many published demonstrations of renewal in female Wistar rats, would be important.

6. PLOS authors have the option to publish the peer review history of their article (what does this mean?). If published, this will include your full peer review and any attached files.

Reviewer #1: No

Reviewer #2: No

---

## [Author Response · Author response to Decision Letter 0]

26 Apr 2022

The response to reviewers is attached.

---

## [Decision Letter · Decision Letter 1]

1 Jun 2022

Sex differences in the immediate extinction deficit and renewal of extinguished fear in rats

PONE-D-22-04660R1

Dear Dr. Maren,

We’re pleased to inform you that your manuscript has been judged scientifically suitable for publication and will be formally accepted for publication once it meets all outstanding technical requirements.

Kind regards,

Fred J. Helmstetter, PhD

Academic Editor

PLOS ONE

Additional Editor Comments (optional):

Reviewers' comments:

Reviewer's Responses to Questions

**Comments to the Author**

1. If the authors have adequately addressed your comments raised in a previous round of review and you feel that this manuscript is now acceptable for publication, you may indicate that here to bypass the “Comments to the Author” section, enter your conflict of interest statement in the “Confidential to Editor” section, and submit your "Accept" recommendation.

Reviewer #1: All comments have been addressed

Reviewer #2: All comments have been addressed

2. Is the manuscript technically sound, and do the data support the conclusions?

Reviewer #1: Yes

Reviewer #2: Yes

3. Has the statistical analysis been performed appropriately and rigorously? 

Reviewer #1: Yes

Reviewer #2: Yes

4. Have the authors made all data underlying the findings in their manuscript fully available?

Reviewer #1: Yes

Reviewer #2: Yes

5. Is the manuscript presented in an intelligible fashion and written in standard English?

Reviewer #1: Yes

Reviewer #2: Yes

6. Review Comments to the Author

Reviewer #1: The authors have addressed all of my initial comments and concerns and I believe the manuscript it now suitable for publication.

Reviewer #2: The authors have addressed all of my concerns from the previous version and I have no further comments.

7. PLOS authors have the option to publish the peer review history of their article (what does this mean?). If published, this will include your full peer review and any attached files.

Reviewer #1: No

Reviewer #2: No

---

## [Editor Report · Acceptance letter]

3 Jun 2022

PONE-D-22-04660R1 

Sex differences in the immediate extinction deficit and renewal of extinguished fear in rats 

Dear Dr. Maren:

I'm pleased to inform you that your manuscript has been deemed suitable for publication in PLOS ONE. Congratulations! Your manuscript is now with our production department. 

Kind regards, 

on behalf of

Dr Fred J. Helmstetter 

Academic Editor

PLOS ONE